# Endovascular Management of Vascular Complications in Ehlers–Danlos Syndrome Type IV

**DOI:** 10.3390/jcm11216344

**Published:** 2022-10-27

**Authors:** Mubarak Alqahtani, Amandine Claudinot, Marine Gaudry, Axel Bartoli, Pierre Antoine Barral, Vincent Vidal, Louis Boyer, Tiffany Busa, Farah Cadour, Alexis Jacquier, Mariangela De Masi, Laurence Bal

**Affiliations:** 1Department of Radiology, Hôpital de la Timone, AP-HM, 13005 Marseille, France; 2Department of Radiology, University Hospital, 14000 Caen, France; 3Aortic Center, Hopital de la Timone, AP-HM, 13005 Marseille, France; 4Department of Vascular Surgery, Hopital de la Timone, AP-HM, 13005 Marseille, France; 5CRMBM-UMR CNRS 7339, Aix-Marseille University, 13007 Marseille, France; 6Department of Radiology, University Hospital, 63100 Clermont-Ferrand, France; 7Department of Medical Genetics, Hopital Enfants de la Timone, AP-HM, 13005 Marseille, France; 8Regional Reference Department for Marfan and Related Diseases, AP-HM, 13005 Marseille, France

**Keywords:** Ehlers–Danlos syndrome, complication, embolization, therapeutic, aortic dissection

## Abstract

(1) Background: The vascular type of Ehlers–Danlos syndrome (vEDS) is a rare genetic connective tissue disorder caused by pathogenic variants in the COL3A1 gene that result in arterial and organ fragility and premature death. We present five cases of vEDS that highlight the diagnosis and treatment challenges encountered by clinicians with these patients. (2) Case presentations: we present the cases of five patients with vascular complications of vEDS who were successfully managed using endovascular interventions or hybrid techniques at our institution from 2005 to 2022. (3) Conclusions: These data emphasize that a multidisciplinary approach is needed for vEDS patients and that when endovascular or hybrid treatment is performed in a timely manner by a skilled team of interventional radiologists, good results can be achieved. Our report also demonstrates that the prognosis of vEDS patients has improved over the past 20 years with a new prevention program including celiprolol therapy, physical activity adaptation and limitation, and scheduled monitoring by expert clinicians.

## 1. Introduction

Ehlers–Danlos syndrome (EDS) is a heterogeneous group of inherited connective tissue disorders characterized by systemic manifestations such as skin fragility and hyperextensibility, joint hypermobility and kyphoscoliosis, and arterial or digestive complications [1,2]. Type IV EDS, also known as vascular EDS (vEDS), was first described by Sack and later modified by Barabas [3], and its prevalence is estimated at 1/50,000 to 200,000 persons [2]. vEDS results from pathogenic variants in the COL3A1 gene, which encodes type III procollagen, a major protein in vessel walls and hollow organs [4,5]. The transmission of vEDS is autosomal dominant with a high penetrance, but genotype–phenotype correlations have been described with more severe and premature vascular risk in the presence of mutations in glycine residues. In addition to vascular catastrophes, a positive family history and specific clinical features characterizing extravascular involvement of Ehlers–Danlos disease can raise the clinical suspicion of vEDS, which is confirmed by genetic testing [5]. Early diagnosis of the disease is important, and patients who benefit from secondary prevention with overall medical care and celiprolol treatment exhibit a lower annual occurrence of arterial complications and a higher survival rate [6]. Celiprolol has been proven to have a protective effect against vascular complications in vEDS [7]. Unfortunately, vEDS is often diagnosed in emergency situations following acute vascular events (rapid aneurismal development and progression, arterial dissection, and aneurysmal ruptures), often with multiple and simultaneous locations. The management of such vascular complications in vEDS patients is challenging due to the tremendous fragility of the vascular tissue during these acute periods. Open surgery and endovascular treatment are associated with high complication rates and a risk of mortality [8]. Small case series have shown lower risks associated with endovascular methods [9,10,11]. Here, we present the cases of five patients with vascular complications of vEDS, which were successfully managed using endovascular interventions or hybrid techniques (Table A1).

## 2. Detailed Case Description

### 2.1. Case 1

A 36-year-old female was referred to our hospital presenting with hemoptysis and respiratory distress. One week before she delivered (35 weeks gestation), she experienced hemorrhagic shock due to the rupture of a splenic artery pseudoaneurysm. She was treated by a cesarean birth and a splenectomy. The patient was managed using mechanical intubation and bilateral pleural drainage. An enhanced thoracic computed tomography (CT) scan revealed an intraparenchymal hemorrhage in the right upper lobe with an arteriovenous fistula between the apical segmental pulmonary artery and the azygos vein, as well as a pseudoaneurysm at the ostium of the middle segmental artery (Figure 1).

The diagnosis of EDS was suspected based on the patient’s clinical history and CT scan findings, and endovascular management was decided with a thoracic surgeon. Under local anesthesia, a 10 F introducer sheath (Radiofocus® introducer II, Terumo Corporation, Tokyo, Japan) was inserted via the right femoral vein. The right pulmonary artery was catheterized using a 6 F ENVOY catheter (Codman-USA), and the distal segment of the arteriovenous fistula was occluded using multiple 0.035 hydrocoils (Figure 2A,B). The ostium of the arteriovenous fistula was occluded with second-generation plugs (AMPLATZER-USA) (Figure 2C), and the pseudoaneurysm was occluded using multiple 3D coils (Figure 2D).

The diagnosis of vEDS was confirmed by the presence of a pathogenic variant in COL3A1 (glycine mutation). After 8 years of follow-up in our reference center for rare vascular diseases, the patient is still alive under celiprolol therapy, despite three other acute arterial dissections (the superior mesenteric, hepatic, and renal arteries) and a celiac trunk rupture requiring embolization.

### 2.2. Case 2

A 25-year-old male patient, with a familial history aortic rupture leading to the sudden death of his father in 1999, was referred to our hospital after the appearance of a left groin hematoma. An enhanced abdominal and pelvic CT scan revealed an aneurysm of the splenic artery measuring 30 × 25 mm and a recent dissection of the left common iliac artery, with a maximum iliac diameter of 12 mm (Figure 3A,B). In his medical history, he complained of abdominal pain within the left hypochondrium for the past 8 months following physical exercise. After a multidisciplinary evaluation, we first decided to embolize the splenic aneurysm using coils to prevent the risk of rupture. Under general anesthesia, a 6 F introducer was inserted into the left humeral artery using the Seldinger technique. Two-dimensional/three-dimensional fusion with preoperative CT angiography was performed to guide the catheter and microcatheter for the more rapid catheterization of the aneurysm and to decrease the radiation exposure (Figure 3C). We catheterized the celiac trunk artery using a 6 F ENVOY catheter (Codman-USA) and then catheterized the aneurysm using a Progreat 2.8 F microcatheter (Terumo-Tokyo, Japan). We started to exclude the outflow of the aneurysm with several Ruby coils (Penumbra-Alameda, CA, USA). We then filled the aneurysm with multiple coils starting with a 3D Ruby standard coil of 34 mm diameter and after decreasing the diameter of the coils using interlock detachable coils (IDCs, Boston Scientific, Marlborough, MA, USA), Nester coils (COOK 6 USA), and packing coils (Penumbra-Alameda, CA, USA), and we occluded the aneurysm inflow using Ruby standard coils (Penumbra-Alameda, CA, USA) (Figure 3D).

Successive CT scan follow-up revealed rapid aneurysmal growth of his left common iliac artery with a 12 mm increase in diameter within one week. The patient complained of left groin pain. A hybrid surgical treatment of left iliofemoral bypass and endovascular exclusion of the internal iliac artery was selected (Figure 4).

During the iliofemoral bypass, a 6 Fr introducer was inserted into the left femoral artery. The left internal iliac artery was catheterized using both a 4 F cobra catheter (COOK-MEDICAL) and a Progreat 2.7 F catheter (Terumo-Tokyo, Japan). The internal iliac artery trunk was occluded using 6 × 60 POD coils (Penumbra-Alameda, CA, USA). Then, the common internal iliac artery was occluded using second-generation 10 mm Amplatzer vascular plugs and several coils (ABBOTT). The external iliac artery was occluded by several Amplatzer Plugs (ABBOTT). One week after the intervention, he complained of acute epigastric pain and presented with hemorrhagic shock. An enhanced abdominal and pelvic CT scan showed hemoperitoneum with aneurismal rupture of the left hepatic artery and an aneurysm of the celiac trunk and common hepatic artery (Figure 5). The patient was referred in extreme emergency to the interventional radiologist. A 6 F femoral access was performed to access the celiac trunk using the 6 F RDC catheter (Cordis-Santa Clara, CA, USA). Then, the left hepatic artery aneurysm was catheterized using a Progreat 2.7 F microcatheter (Terumo-Tokyo, Japan). The rupture of the left hepatic artery was confirmed on digital subtraction angiography (DSA), and the vessel was occluded using cyanoacrylate glue and multiple coils. DSA of the celiac trunk revealed an aneurismal evolution of the celiac trunk and the common hepatic artery with an occlusion of the common hepatic artery before the gastroduodenal artery. The common hepatic artery and celiac trunk were occluded using ethylene vinyl alcohol copolymer (Onyx, ev3, USA), several interlocking coils, and IDS (Boston Scientific, Marlborough, MA, USA) (Figure 5) without any complications during the procedure.

The patient’s family history, clinical features (translucent skin, facial morphotype), and acute vascular storm with successive dissections led to the clinical diagnosis of vEDS, which was confirmed by the identification of a Gly mutation in the COL3A1 gene. The patient is alive in good health after 2 years of follow-up under celiprolol therapy and without any vascular event.

### 2.3. Case 3

A 25-year-old female patient with pre-symptomatic vEDS was referred to our hospital for acute left-sided hemiplegia and transient right arm hemiparesis. Magnetic resonance imaging (MRI) cerebral and CT angiography of the supra-aortic trunk demonstrated a sub-occlusion of the right middle cerebral artery, along with a severe hypoperfusion of the right internal carotid artery related to dissection and an intramural hematoma of the left internal carotid artery. Her medical status was a contraindication to thrombolysis, and the stroke team decided to refer her to the interventional neuroradiologist. Under general anesthesia, an 8 F introducer was inserted via the right femoral artery. Right common carotid artery access was obtained using a 7 F NEURON MAX catheter (Penumbra-Alameda, CA, USA) and a TERUMO 0.035 guidewire. DSA revealed an occlusion of the dissected right internal carotid artery and an occlusion of the proximal portion of the middle cerebral artery (M1) (Figure 6A). Aspiration through a dedicated catheter (Sofia 6 F TERUMO) of the M1 thrombus was unsuccessful (Figure 6B,C). The thrombus was then successfully removed using a Headway microcatheter (0.027), Terumo microwire (0.016), and stent retriever (Solitaire 4 × 20), with good outcome (Figure 6D). The right internal carotid artery dissection was treated through the deployment of a 7 × 50 mm carotid wall stent in the extracranial portion of the right internal carotid artery (Figure 6E,F).

No complications occurred during the procedure or during the postprocedural hospital stay. A few months later, she developed a dissecting aneurysm in the right internal carotid artery after the stent, which was monitored by Doppler ultrasound. After 3 years of follow-up, she did not have any other arterial event under celiprolol therapy and presented with persistent left hemiparesis.

### 2.4. Case 4

A 43-year-old female patient with a clinical and genetic diagnosis of vEDS (Gly mutation) visited our emergency room in a state of hemorrhagic shock. An enhanced abdominal CT scan showed a retroperitoneal hematoma surrounding the right kidney, with a ruptured aneurysm of the right renal artery. The patient was referred to the interventional radiologist. Under local anesthesia, the right renal artery was reached using a 6 F RDC catheter (Cordis-Santa Clara, CA, USA), and the right renal artery was then embolized using multiple IDCs and interlocking coils (Figure 7) without any complications during the procedure.

After 12 years of follow-up, she is still alive under medical treatment with celiprolol and losartan, despite two acute symptomatic arterial dissections (the superior mesenteric and gastric arteries) managed by a medical approach, and regular monitoring revealed an asymptomatic right middle cerebral artery aneurysm and splenic aneurysm.

### 2.5. Case 5

A 35-year-old male patient was admitted to the emergency department for acute abdominal pain. The CT scan showed a celiac trunk dissection extended to the hepatic and splenic arteries, a left renal artery dissection with an infarct and a right common iliac artery dissection with an aneurysmal formation (Figure 8A). The patient was monitored in the intensive care unit and treated with antihypertensive and analgesic anticoagulation therapy and strict rest. His clinical phenotype indicated vEDS according to the vascular medical team, and celiprolol was introduced. Four days later, he presented with recurrent abdominal pain associated with severe lumbar pain. A CT scan demonstrated a rapid diameter evolution of the left renal artery aneurysm and right common iliac artery dissection and a new dissection of the left common iliac and right renal arteries (Figure 8B). The pain regressed with blood pressure control and analgesics. A clinical and radiological follow-up was decided. One week later, due to the recurrence of lumbar and abdominal pain and aneurysmal evolution on two common iliac arteries, the patient was referred to a vascular surgeon and interventional radiologist. Under general anesthesia, we performed left hypogastric embolization by coils and endovascular exclusion of the left iliac artery with an aorto-iliac stent graft (Medtronic ETLW 16-10C124EE and ETEW 20-20C82EE). The contralateral iliac artery was embolized with second-generation 10 mm plugs (AMPLATZER-USA), and we performed a femoro-femoral bypass from the left to the right (Figure 8C).

CT scan follow-up demonstrated an increase in the dissection of the left renal artery aneurysm, and multidisciplinary teams decided to refer the patient to interventional radiology for the embolization of the left retropyelic artery and prepyelic artery stenting. Under local anesthesia, a 6 F introducer was inserted into the left femoral artery, and the left renal artery was reached using a 5 F cobra catheter (COOK-MEDICAL). The left retropyelic artery was embolized with coils and a small amount of ethylene vinyl alcohol copolymer (Onyx, ev3, USA). Then, a two carotid wall stent (Boston Scientific, Marlborough, MA, USA) was deployed, 5 mm × 30 mm and 7 mm × 30 mm, in the prepyelic artery without any complications from the procedure (Figure 9).

The patient returned to intensive care for a few days. Then, he was monitored in the vascular surgery department, and the immediate postoperative period was uneventful. The patient was discharged at 30 days post operation. The patient was in good health after 10 months of follow-up by our vascular team, medication therapy with celiprolol and irbesartan, and physical activity adaptation. A regular to invasive progression has been observed thus far. The diagnosis of vEDS was confirmed by the identification of a pathogenic Gly variant in the COL3A gene.

## 3. Discussion

Genetic analysis is the reference method to confirm a clinical vEDS diagnosis, but it is not performed in emergency situations. Therefore, the detection of clinico-radiological patterns compatible with the diagnosis is crucial to manage these patients and to discuss the therapeutic options with a vascular team in a reference center (a vascular specialist and a surgeon, radiologist, geneticist, and anesthesiologist) [12,13]. The first-line treatment of vascular complications should be medical stabilization of blood pressure in the normal range [11,14] and clinical and imaging monitoring during a strict rest period of hospitalization. Frank et al. demonstrated in a recent observational study that celiprolol treatment improved survival in patients with vEDS [6].

The management of such vascular complications in vEDS patients is challenging. Due to the fragility of the arterial wall, open surgery and endovascular treatment have high complication rates and risks of mortality [9,10]. Freeman et al. reported major complications in 22% of cases and a death rate of 5.6% among 18 vEDS patients who underwent angiography [15].

Schievink et al. reported that 24% of 25 angiographies resulted in severe bleeding at the arterial puncture site with manual compression [16]. However, Okada T et al. recently reported fewer complications, reporting that three minor complications occurred in seven procedures [11,17,18]. A recent report showed more encouraging results, probably due to medical progress over time. Linfante et al. presented a case of successful management of a carotid cavernous fistula in a patient with confirmed vEDS [19]. In another case of a ruptured subclavian artery aneurysm in a confirmed case of vEDS, Lida et al. described how they managed the case successfully with transcatheter coil embolization [20]. In our case series, only one minor complication occurred at the level of the puncture site, without any deaths.

Concerning therapeutic strategy, the cornerstone of the treatment decision is, in our opinion, a multidisciplinary approach in a reference center for rare vascular diseases. These cases must be discussed collegially during in-person meetings including vascular physicians, vascular and cardiac surgeons, and interventional and diagnostic radiologists. For emergency cases, this multidisciplinary team must be able to consult rapidly around the clock with the on-call specialists to quickly discuss the best strategy to adopt.

Based on a recent publication and the presented data, the decision to treat should be made before vascular rupture and based on the speed of vascular growth and the ratio between the size of the native artery that holds the aneurysm and the aneurismal diameter itself. From our point of view, the decision to treat should not be postponed because the patient is suspected to have vEDS. Pregnancy is suspected to promote vascular storms in such patients, and vEDS has one of the highest mortality rates during labor or in the postpartum period. A recent report in the literature supports delivery by cesarean section at 32 weeks [21].

During the acute storm phase, a new aneurismal location appears and increases in size. The right time to treat these lesions during the follow-up period should also be discussed. For instance, in Case 2, the iterative embolization and the hybrid intervention might have enhanced the systemic vascular stress and inflammation and the occurrence of hepatic artery rupture during hospitalization.

The treatment option could be an endovascular [11] hybrid or surgical approach, depending on the features of each case.

Several recently reported cases with aneurysmal rupture in vEDS patients were managed successfully with endovascular procedures [10,19,20]. There are currently no international recommendations on the management of these visceral aneurysms, either in an emergency or a scheduled procedure. The two main series of patients from the Mayo Clinic and Johns Hopkins are now more than 10 years old. The Mayo Clinic study showed a higher perioperative mortality for open surgical treatment. However, these surgeries were mostly performed on the aorta, and few patients were treated for visceral aneurysms [22]. In their literature review of vEDS, Bergqvist et al. had a higher mortality rate with open surgical repair of arterial complications (30%) than with endovascular procedures (24%) [10]. Concerning open surgical treatments, recommendations have been made but only from clinical observations. The use of vascular clamps or balloon occlusion is not recommended, and careful suture with pledget support or graft covering should be favored [23].

In our opinion, endovascular treatments should be preferred for the treatment of visceral aneurysms in vEDS. Endovascular treatment can provide quick, feasible, and safe treatments for these complications, especially in emergency situations. However, physicians should be aware that risks remain with these procedures, and operators should pay attention to arterial access complications, catheter and guidewire damage, and the periprocedural risk of rupture. Procedures should be carried out under close anesthetic monitoring, with balloon occlusion and, most importantly, liquid embolization devices quickly accessible in these situations. Okada et al. presented the results of seven successful emergency endovascular treatments of vEDS, with only minor complications [13].

All these patients are still alive today. With such a rare disease, it seems impossible to establish the real mortality based on these cases. The cases of all patients who were treated for this disease in the angiography department of our institute are presented in this article. Two patients known from our center for infracentimetric visceral aneurysms died outside the institution—one from a cardiac complication and one from an unknown cause (no follow-up). We cannot exclude the possibility of an aneurysmal rupture.

## 4. Conclusions

These data underline that a multidisciplinary approach is needed for vEDS patients and that when endovascular or hybrid treatment is performed in a timely manner by a skilled team of interventional radiologists, good results can be achieved. Our report also demonstrates that the prognosis of vEDS patients has improved during the past 20 years with a new prevention program including celiprolol therapy, physical activity adaptation and limitation, and scheduled monitoring by expert clinicians.

## Figures and Tables

**Figure 1 jcm-11-06344-f001:**
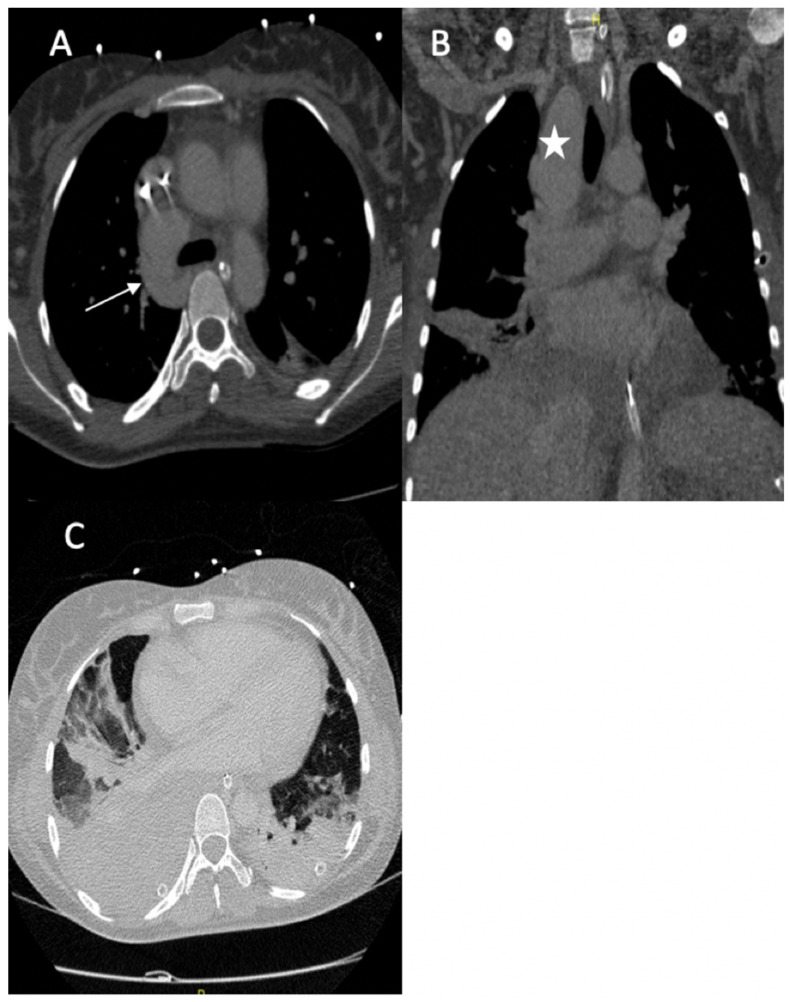
(**A**) Axial image from the contrast-enhanced thoracic CT scan revealed an arteriovenous fistula between the apical segmental artery and the azygos vein (arrow); (**B**) coronal section of the CT scan revealed a pseudoaneurysm of the right lobe superior artery (star); (**C**) axial image thoracic CT scan (pulmonary window) showed bilateral pleural effusions with ground glass opacity, most likely due to pulmonary hemorrhaging.

**Figure 2 jcm-11-06344-f002:**
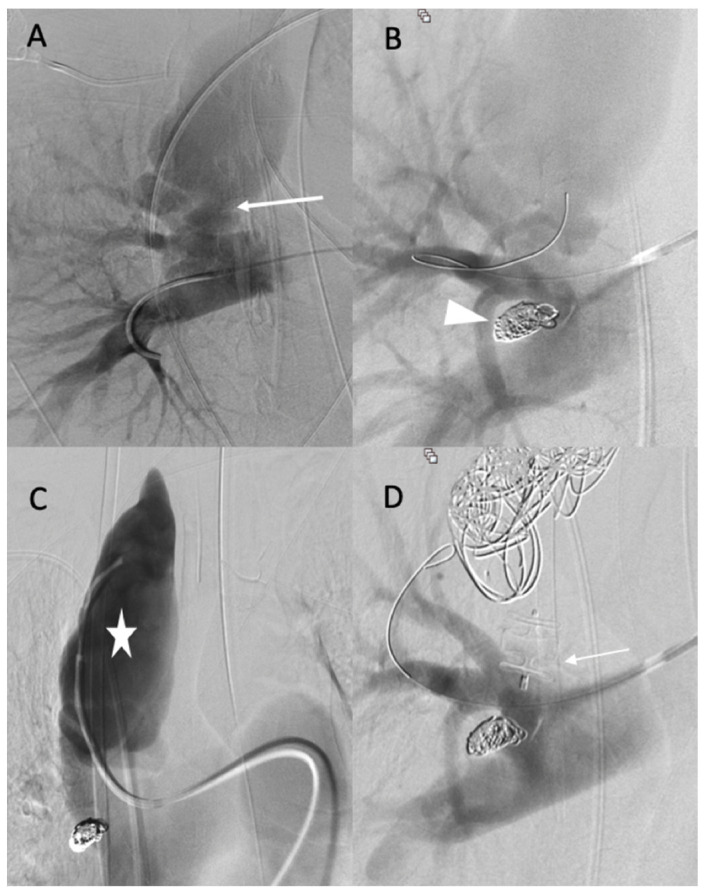
(**A**,**B**) Before and after the occlusion of the distal portion of the arteriovenous fistula (arrow, head arrow). (**C**) Before the occlusion of pseudoaneurysm (star). (**D**) The occlusion of the proximal portion using first- and second-generation plugs (arrow), and the occlusion of the pseudoaneurysm using multiple 3D coils.

**Figure 3 jcm-11-06344-f003:**
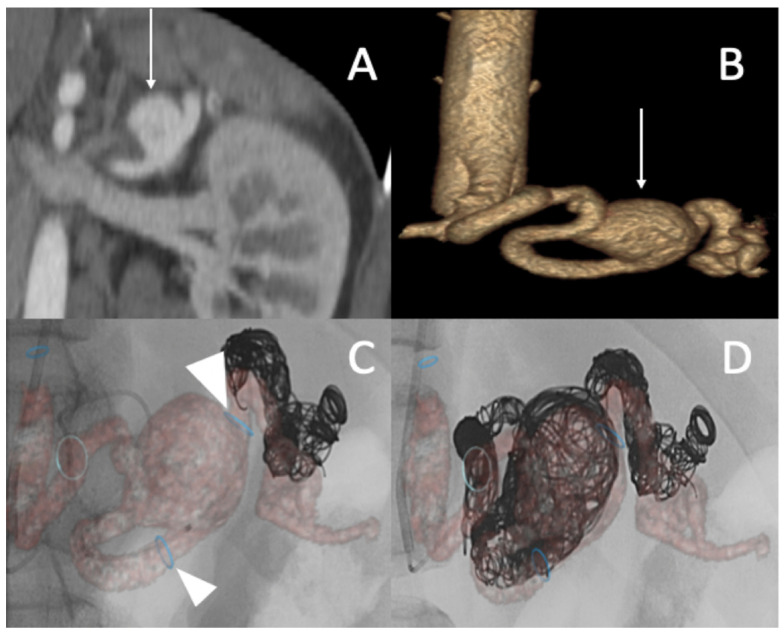
(**A**,**B**) Abdominal CT angiogram with 3D image and arteriography with image fusion (**C**,**D**) showing the aneurysm in the splenic artery (arrows). We marked the proximal and distal extremities of the neck of the aneurysm (head arrow), and then we started the embolization using multiple coils after, within, and before the aneurysm.

**Figure 4 jcm-11-06344-f004:**
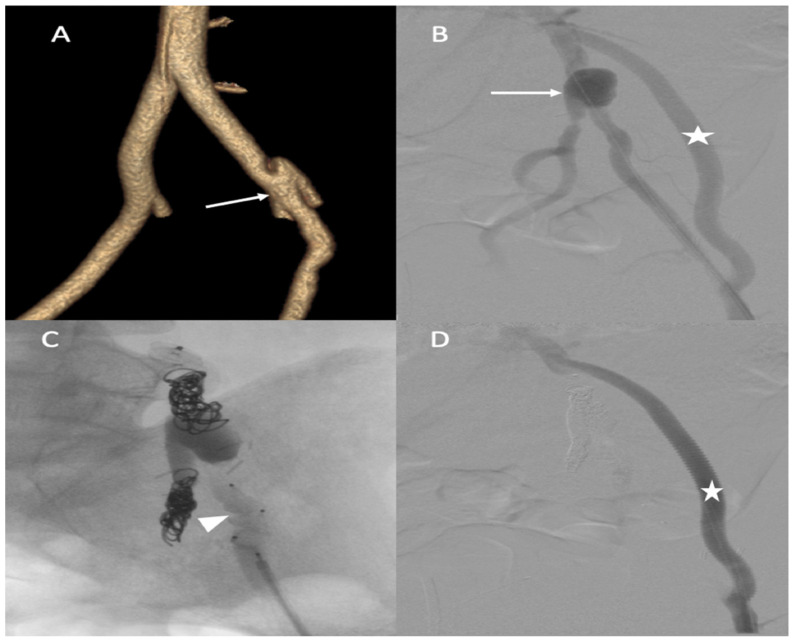
(**A**) Abdominal CT angiogram with 3D image showing the dissection of the left common iliac artery (arrow); (**B**) arteriography shows the left iliofemoral bypass (star) and dissection of the left common iliac artery (arrow); (**C**) occlusion of the left internal and external iliac arteries by coils and then a plug (head arrow); (**D**) arteriography final shows the permeability of left iliofemoral bypass (star).

**Figure 5 jcm-11-06344-f005:**
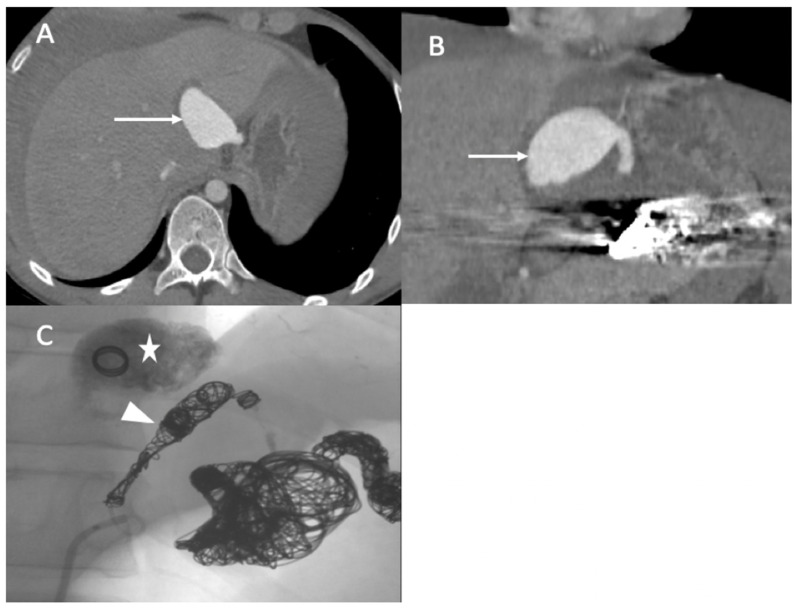
(**A**,**B**) Axial and coronal abdominal CT images pertaining to the arterial phase show the aneurysm in the left hepatic artery (arrows) with perihepatic hematoma; (**C**) one coil was inserted in the aneurysm (star), and then glue and multiple coils were introduced into the common hepatic artery (arrowhead).

**Figure 6 jcm-11-06344-f006:**
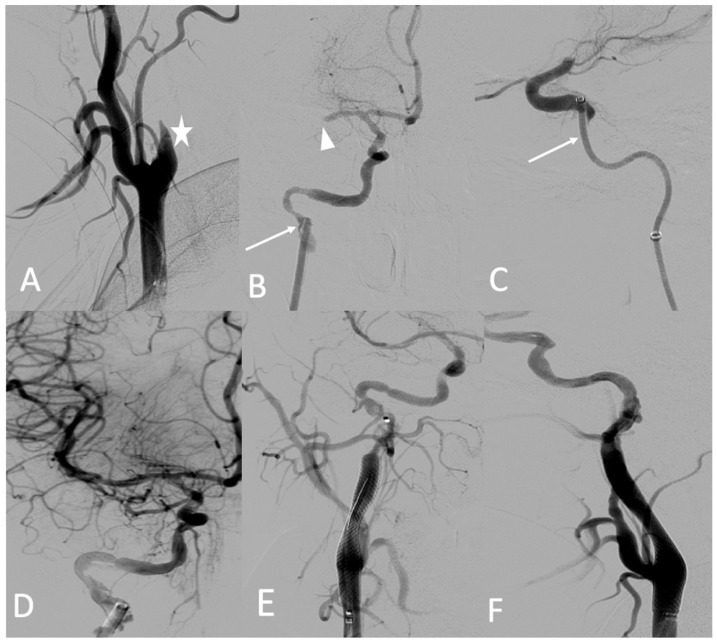
(**A**) Digitally subtracted image from the right common carotid angiogram showing occlusion over the dissection of the right internal carotid artery (star); (**B**,**C**) crossing the dissection by a Sofia 6 F aspiration catheter (arrows) and the occlusion of the proximal portion (M1) (arrowhead); (**D**) anterior view after mechanical thrombectomy; (**E**,**F**) anterior and lateral views after the deployment of the carotid wall stent.

**Figure 7 jcm-11-06344-f007:**
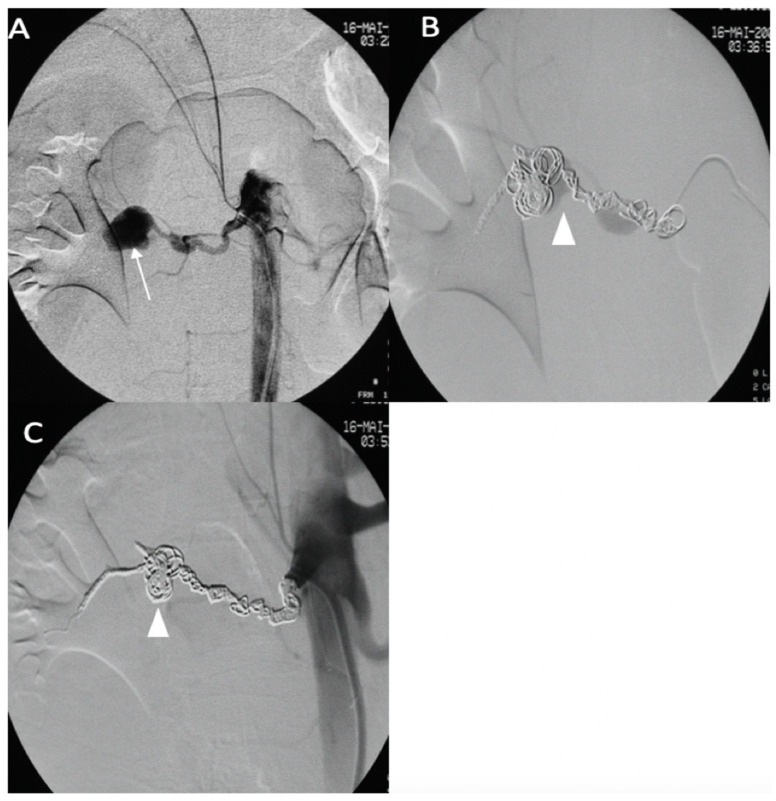
(**A**) Digitally subtracted image from the right renal artery revealing the distal aneurysm with irregular walls (arrow). (**B**,**C**) Postembolization by coils.

**Figure 8 jcm-11-06344-f008:**
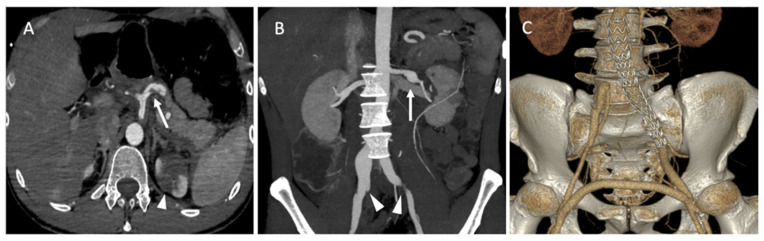
(**A**) Axial abdominal CT images in the arterial phase showing a celiac trunk dissection extended to the hepatic and splenic arteries (arrows) with renal ischemia; (**B**) coronal abdomen and pelvis CT images in the arterial phase showing common right and left iliac artery dissection (arrowheads); (**C**) the abdominal angiogram CT with 3D imaging reveals the left hypogastric embolization by coils and the endovascular exclusion of the left iliac artery with an aorto-iliac end prosthesis. The contralateral iliac artery was embolized with second-generation plugs and femoro-femoral bypass from left to right.

**Figure 9 jcm-11-06344-f009:**
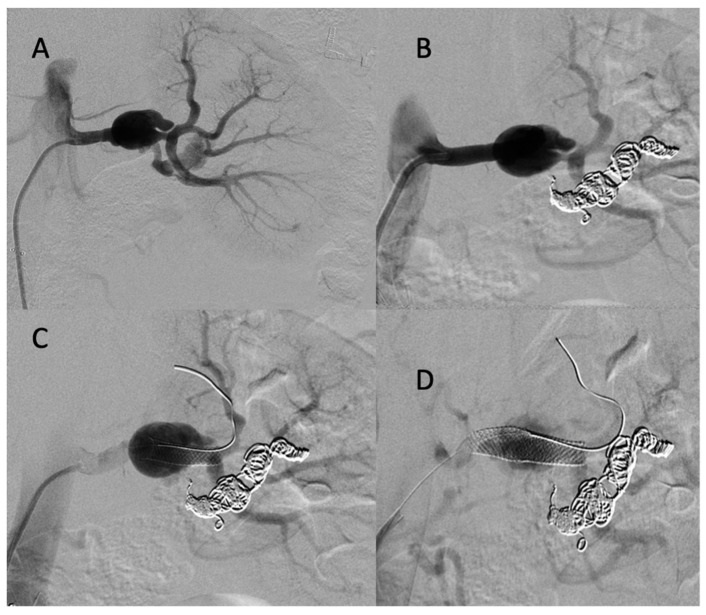
(**A**) Digitally subtracted image from the left renal artery dissection with aneurysm formation; (**B**) postembolization of the left retropyelic artery by multiple coils and a small amount of ethylene vinyl alcohol copolymer; (**C**,**D**) post-deployment of a two carotid wall stent in the left prepyelic artery.

## Data Availability

The data supporting the findings of this study are available within the article.

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
