# Peer review of "Endovascular Management of Vascular Complications in Ehlers–Danlos Syndrome Type IV"

_jcm, 2022, doi:10.3390/jcm11216344_

Round 1

Reviewer 1 Report

The authors are to be congratulated for well written manuscript, but there a few minor suggestions:

1) Celiprolol is a unique beta blocker a would be valuable addition to the introduction

2) The authors conclude that the best approach to these emergent case is a multidisciplinary approach.  How was this approached acted on?  Where their case conferences or zoom meetings?

3) Finally the authors need to comment on selection bias.  The five cases presented all did well and this may not be realistic.

Author Response

The authors are to be congratulated for well written manuscript, but there a few minor suggestions:

Response: Thank you for your comments and revisions.

  • Celiprolol is a unique beta blocker a would be valuable addition to the introduction

Response: Following your advice, modifications have been made in the revised version of the manuscript.

  • The authors conclude that the best approach to these emergent case is a multidisciplinary approach.  How was this approached acted on?  Where their case conferences or zoom meetings?

Response: All these cases are discussed collegially at in-person weekly meetings including vascular physicians, vascular surgeons, cardiac surgeons, interventionnal and vascular radiologists.  These meetings allow to present challenging cases, and choose the best  therapeutic strategy. Concerning emergency or deep night cases, our structure benefits from a dedicated Aortic Center organization with 24/7 on-call surgeons and vascular interventional radiologists that can collectively decide the best strategy.

  • Finally the authors need to comment on selection bias.  The five cases presented all did well and this may not be realistic.

Response: It is difficult to answer to that question. Some of these patients required a very close follow up due to recurrent vascular complication in several territory. For some reason complication stops after a while without any obvious reasons. Patient with such disease should be recovered in intensive care unit when the first vascular complication arise and followed up clinicaly and with CT scan very closely.

All these patients are still alive today this is impossible with such a rare disease to establish the real incidence of mortality based on these cases. All patients that have been treated in the angiography room for this disease are presented in this article. 2 patients, known from our center for infracentimetric visceral aneurysms have died outside the institution. One from cardiac complication. One from unkown cause (no follow-up). We cannot exclude an aneurysmal rupture.

Reviewer 2 Report

I think that this is a superbly written and illustrated paper.  There is still a lot of equipoise about endovascular management particularly of aneurysms in patients with her eligible congenital aortic disease including Ehlers-Danlos.  Some sinus report excellent results with open repair depending on the setting.  A lot of it depends on how long the follow-up is.  There is still concern of the long-term durability of endovascular solutions particularly stenting for aneurysms.  However what the authors illustrate and really I think the main point of the paper is that it takes a specialized multidisciplinary approach.  Endovascular solutions are part of the surgical armamentarium.  A number of these patients really did not have good open surgical options.

I do not think that the paper says that in every case the approach that the team used was the correct 1 technically but it certainly had good results at least an interim follow-up and more importantly rather less than the specific approach with the team approach by a group of experts that is the point worth emphasizing

Author Response

Thank you for your comments and revisions.

Editor comments:

Thank you for submitting your manuscript. I found the 5 case reports extremely interesting, and the topic of vascular complications in EDS, especially those involving visceral vessels, is worthy of consideration for our Special Issue. However, the discussion is too brief, and it should be more focused on the different techniques that could be used and their risults, especially in the treatment of visceral arteries.
Moreover, a careful English editing is needed bifore the manuscript can be accepted.
I am looking forward to receive the revised paper, in order to submit it to our reviewers with the utmost dispatch.
Best regards,
Luigi Federico Rinaldi

Response: Thank you for your comments and revisions. Following your advices, discussion has been expanded in the revised version of the manuscript. Also, English editing has been done.